# Parking Space Recognition Method Based on Parking Space Feature Construction in the Scene of Autonomous Valet Parking

**Shidian Ma** [1]**, Weifeng Fang** [1]**, Haobin Jiang** [2]**, Mu Han** [3,*] **and Chenxu Li** [2]

[1] Automotive Engineering Research Institute, Jiangsu University, Zhenjiang 212013, China; masd@ujs.edu.cn (S.M.); 2221804071@stmail.ujs.edu.cn (W.F.)
[2] School of Automotive and Traffic Engineering, Jiangsu University, Zhenjiang 212013, China; Jianghb@ujs.edu.cn (H.J.); 1000005020@ujs.edu.cn (C.L.)
[3] School of Computer Science and Communication Engineering, Jiangsu University, Zhenjiang 212013, China
* Correspondence: hanmu@ujs.edu.cn

**Abstract:** At present, the realization of autonomous valet parking (AVP) technology does not achieve information interaction between the parking spaces and vehicles, and accurate parking spaces information perception cannot be obtained when the accuracy of the search is not precise. In addition, when using the camera vision to identify the parking spaces, traditional parking space features such as parking lines and parking angles recognition are susceptible to light and environment. Especially when the vehicle nearby partially occupies the parking space to be parked, it is not easy to determine whether it is a valid empty parking space. This paper proposes a parking space recognition method based on parking space features in the scene of AVP. By constructing the multi-dimensional features containing the parking space information, the cameras are used to extract features' contour, locate features' position and recognize features. In this paper, a new similarity calculation formula is proposed to recognize the stained features through template matching algorithm. According to the relative position relationship between the feature and parking space, the identification of effective empty parking spaces and their boundaries is realized. The experimental results show that compared with the recognition of traditional parking lines and parking angles, this method can identify effective empty parking spaces even when the light conditions are complex and the parking spaces are partially occupied by adjacent vehicles, which simplifies the recognition algorithm and improves the reliability of the parking spaces identification.

**Keywords:** autonomous valet parking; parking space feature; parking space recognition; machine vision; template matching algorithm

## 1. Introduction

With the scientific and technological progress, technologies related to unmanned driving have become a research hotspot in the field of intelligent vehicle technology (IVT). Autonomous valet parking (AVP) can effectively save people's driving time, reduce the difficulty of parking, and greatly improve the utilization rate of empty parking spaces in the parking lot, helping to realize the "last kilometer" task of unmanned driving. Therefore, research and development of AVP systems are of essential significance. Many efforts have been undertaken widely by scholars at home and abroad, resulting in much attention directed towards intelligent automobile manufacturers.

The technology of AVP is mainly realized by the vehicle's automatic driving function, the comprehensive regulation of the parking lot management system and the communication technology between the vehicle and the parking lot [1]. After the parking lot management system senses the real-time status information of the parking space in the field, it sends the corresponding parking space and recommended driving path information to the vehicle through the communication network. Then the vehicle automatically moves to the designated parking space according to the recommended path for parking. In the

current research and application, the AVP technology is mainly realized through in-vehicle and infrastructure-based system. Urmson and Dolgov [2,3] realized the AVP function of multi-level parking structure through the on-board laser scanner and inertial measurement unit (IMU). Jeevan [4] proposed a method to realize AVP through camera and odometer positioning with automotive grade sensors. Min [5] finished vehicle positioning in parking lot only through visual information by use of implementing an intelligent vehicle system for AVP service in low speed limited parking area. Klemm and Furgale [6,7] made use of cameras and ultrasonic radar sensors to navigate in the parking lot without using global position system (GPS) to realize the AVP function. Schwesinger [8] just relied on cameras and ultrasonic sensors to complete AVP, which pushed visual localization, environment perception and automated parking to centimeter precision. Ma [9] realized the identification of the vertical parking scene by using machine vision and pattern recognition technology to improve the utilization of parking spaces and the convenience of parking. The implementation above-mentioned requires high-accuracy sensors. In parking space recognition, ultrasonic radar sensors depend much on obstacles on both sides of the parking space. The camera cannot detect worn parking lines, especially in dimly lit condition. Moreover, the recognition accuracy of the parking space boundary is not precise, and accurate parking space perception can not be achieved.

The infrastructure-based system has evolved from a simple parking guidance system in the early stage to a system relying on high-precision sensors installed in the parking lot. The vehicle only needs the systems of electronic braking, automatic gear shifting, electronic power steering and remote inter-connection functions. Min [10] proposed a hierarchical parking guidance model based on roadside guidance display, which can help the driver choose parking spaces and remotely monitor the states of the vehicle. Liu [11] devised a model of a large-scale parking lot guidance information system that used ant colony optimization algorithm to complete the path guidance in the parking lot. Yue [12] utilized ZigBee network and geomagnetic sensors to realize an intelligent parking guidance system to complete AVP. Wang [13] developed a new system for intelligent navigation system used in parking lots. After he had modeled the parking lot with a time-varying graph, the proposed system was applied in a time-varying shortest path algorithm and dynamically tuned arc transit times based on planned vehicle routes, as well as traffic flow sensor data. Zhang [14] studied two-dimensional lidar to construct high-precision maps of parking lots to realize accurate real-time positioning and mapping. Zou [15] finished an indoor positioning system based on ultra-wideband, which meets the needs of indoor positioning systems in most scenarios. The indoor positioning accuracy of the system can reach 10 cm. Although the strategy above-mentioned not only achieves the communication between parking lot and vehicle, but also between parking lot and parking space, it does not fulfill the interaction between parking space and vehicle. So the closed-loop communication is not formed. Moreover, the construction and maintenance of infrastructure transformation cost a lot. So it is not easy to promote on a large scale.

In order to solve these challenges, a parking space recognition method based on parking space features construction in the scene of AVP is proposed in this paper. Firstly, because of the characteristics of low-cost, easy identification and large amount of stored information of the Quick Response (QR) code, it can be used as an information carrier and a new parking space feature that can be recognized by the vehicle to meet the needs of information interaction between parking spaces and vehicles. According to the characteristics of the fisheye camera and the installation location, the arrangement of the QR code on the parking space is proposed to improve the recognition efficiency. Secondly, the parking space images collected by the on-board camera were pre-processed. Since vehicles near the parking space may cross the parking line and the parking space is partially occupied, a method is designed to identify effective empty parking spaces. Thirdly, the template matching algorithm is used to match the QR code which is easy to get defaced. According to the characteristics of the standard binary matrix, a new similarity calculation formula is proposed to improve matching efficiency and accuracy. After the parking space feature

is identified and located by the visual sensor, the parking space and its boundary is recognized according to the relative position relationship between the constructed feature and the parking space. This method simplifies the recognition algorithm and improves the reliability of the parking space recognition. Finally, the actual vehicle experiment verifies the feasibility and effectiveness of this method.

## 2. Parking Space Feature Construction

Typical parking space features include parking lines and parking angles. The current implementation of AVP does not consider the communication between parking spaces and vehicles. If the accuracy of the search the parking space is not precise, it is easy to make vehicles park in the wrong parking space. In order to fix these problems, it is necessary to give the parking space a new feature, which contains multi-dimensional information, and the vehicle interacts with the parking space by locating and identifying the feature. In the scene of AVP, the parking spaces are regular. Therefore, the regular reform of each fixed size parking space is carried out, and the unique QR code with parking number, parking size pasted on the fixed position of each parking space is constructed as the new parking space feature. The multi-dimensional information includes not only the stored data information including the parking space number and the size of the parking space, but also the information of the parking space position that can be obtained when locating the feature QR code. Besides, it also includes information for judging whether the allocated parking space is a valid empty parking space based on the outline of the feature QR code.

### 2.1. The Concept of QR Code

The QR code is a kind of symbol that records data and information in a black and white pattern distributed on a plane (in a two-dimensional direction) with a certain geometric pattern according to a certain rule [16]. The types of QR codes are divided into row-based QR codes and matrix QR codes. The current popular use of QR code belongs to matrix QR code. It is widely used in various fields due to its low cost and large amount of stored information, wide coding range, strong anti-corruption ability and so on, which greatly facilitates people's daily life. With the development of science and technology, QR code technology is also applied in some scientific research fields. Chow [17] presented two methods of verifying the visual fidelity of image-based datasets by using QR codes to detect perturbations in the data, which improved the performance of the machine learning model. Yu [18] combined the binocular vision and QR code identification techniques together to improve the robot positioning and navigation accuracies, and then constructed an autonomous library robot for high-precision book accessing and returning operations.

### 2.2. The Arrangement of QR Code

In this paper, the parking number, parking size and parking charge information are encoded. The QR code selects version 2, which contains 20 characters. The coding error correction level is H, and 30% of the data can be corrected. The encoding of a two-dimensional code refers to the uniform conversion of various characters such as numbers, letters, and Chinese characters into a binary number sequence composed of 0 and 1. The error correction codeword is used to correct the errors caused by the damage of the QR code, and the principle is to perform error correction through the Reed-Solomon algorithm [19]. In order to ensure that the vehicle can recognize the constructed features in real time, it is necessary to obtain panoramic images around the vehicle. This paper uses fisheye cameras on the center of the front and rear bumpers of the vehicle, also on the left and right side of the mirror. The image captured by fisheye lens will produce distortion, and the closer to the center of the image, the smaller the distortion. At the same time, we consider the situation where the vehicles on both sides of the vertical parking space cross the parking lines in the scene of AVP, the parking space size does not meet the parking conditions. This paper sets the size of the QR code to the length of the vertical parking space and pastes it in the middle of the parking space.

## 3. Image Preprocessing

The fisheye lens is a kind of ultra wide angle lens, which is much larger than that of ordinary camera. The effective viewing angle range is close to 180 degrees. The wide imaging range will also produce a big super-wide-angle lens distortion, resulting in obvious distortion of the image [20]. Therefore, it is necessary to calibrate the fisheye camera to correct the image distortion. The calibration of the cameras mainly involves the transformation of four coordinate systems: world coordinate system, camera coordinate system, image plane coordinate system, and image pixel [21]. Zhengyou Zhang calibration method, the radial alignment constraint (RAC) two-stage method of Tsai, and the direct linear transform (DLT) method belong to threes of the calibration methods [22–25].

The main idea of Zhengyou Zhang calibration is to use the black and white checkerboard as the standard template, determine the distortion correction algorithm according to the corner relationship to calculate the internal parameter matrices, distortion coefficient, the rotation matrix R and translation matrix T. The effect of distortion correction is shown in Figure 1.

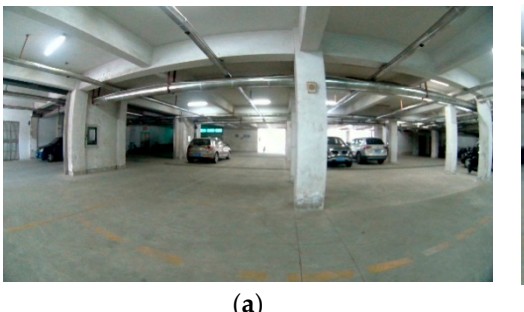 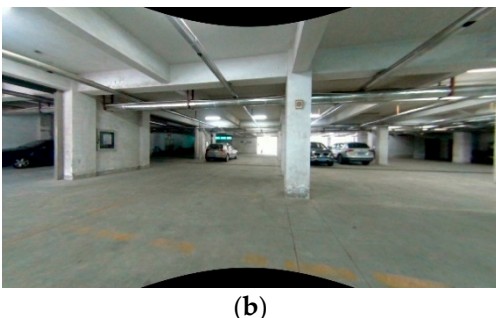

(**a**)        (**b**)

**Figure 1.** The distortion correction of the image. (**a**) The original image; (**b**) the corrected image.

As shown in Figure 2, in order to facilitate the correct identification of parking spaces in a complex environment, the image is converted into a top view through the inverse perspective transformation algorithm. The basic realization principle is to project the initial image imaging plane to another imaging plane, which can be expressed as:

$$\begin{bmatrix} u \\ v \\ 1 \end{bmatrix} = sH \begin{bmatrix} X \\ Y \\ 1 \end{bmatrix} \tag{1}$$

where vector $\begin{bmatrix} u & v & 1 \end{bmatrix}^{\text{T}}$ is homogeneous coordinates of the image after inverse perspective transformation; $s$ is zoom scale; $H$ is image homograph matrix; $[X \, Y]$ is the point in the ground coordinate system.

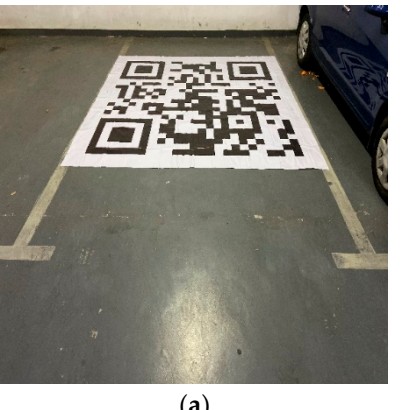 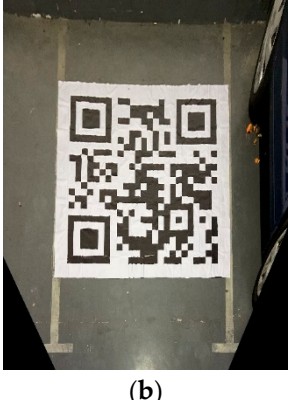

(**a**)        (**b**)

**Figure 2.** Inverse perspective transformation. (**a**) The corrected image; (**b**) top view.

Since the image with the QR code captured by the camera is easily disturbed by noise and environmental problems, the image needs to be preprocessed. These steps include image grayscale, binarization, and filtering.

Image grayscale is to change the three-channel RGB color image into a single-channel grayscale image, which can improve the image processing speed. The color image can be converted to grayscale image by using the following formula [26]:

$$F(i,j) = 0.299 \times R(i,j) + 0.587 \times G(i,j) + 0.114 \times B(i,j) \tag{2}$$

Binarization adopts the binarization algorithm of adaptive threshold. The purpose is to present the image with obvious black and white effect, reduce the amount of data, and highlight the contour of the target. The basic principle is to calculate the local threshold according to the brightness distribution of different areas of the image so as to reduce the impact of lighting conditions, and set the image pixel grayscale value according to the calculated local threshold, which can be expressed as:

$$f(x,y) = \begin{cases} 255 \; if \; g(x,y) > T(x,y) \\ \quad\; 0 \;\; otherwise \end{cases} \tag{3}$$

where $f(x,y)$ is an image pixel grayscale which is set, $g(x,y)$ is a pixel value of the original image, $T(x,y)$ is a threshold calculated individually for each pixel, which can be expressed as:

$$T(x,y) = \frac{1}{M} \sum_{(x,y) \in A} h(x,y) - C \tag{4}$$

where $M$ is the size of a pixel neighborhood, $A$ is the collection of neighborhood point which includes pixels $(x,y)$, $h(x,y)$ is a pixel value of the original image, $C$ is an offset adjustment value. Normally, $C$ is positive but may be zero or negative as well.

In this paper, the median filtering is adopted to smooth and denoise, which can help eliminate the sharp noise of images and realize image smoothing. The basic principle is to take a matrix of pixels in the image, sort by size, and replace the median value of the sorted pixel with the pixel value at the center of the pixel matrix. The result of the image preprocessing is shown in Figure 3. The median filtering calculation formula is proposed as follows:

$$y(n) = med[x(n-N) \cdots x(n) \cdots x(n+N)] \tag{5}$$

where $x(n-N) \cdots x(n) \cdots x(n+N)$ are pixels to be processed, $med[\;]$ means to arrange the values according to the size and then take the middle value.

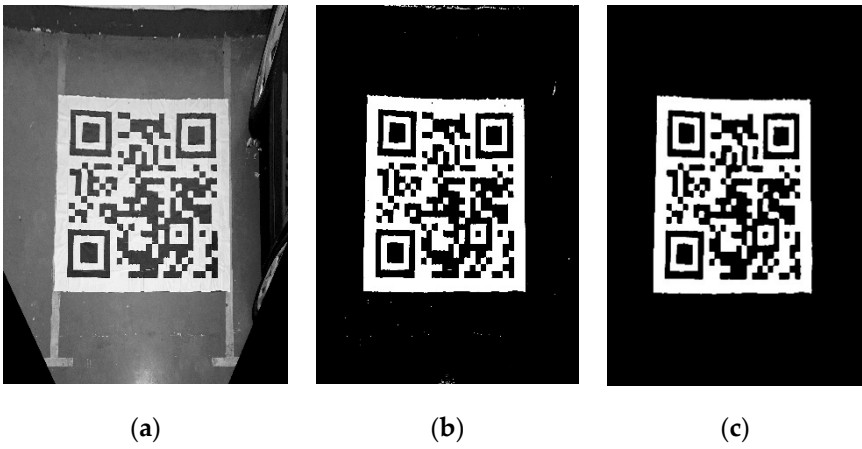

(**a**)　　　　　　　　　(**b**)　　　　　　　　　(**c**)

**Figure 3.** Image preprocessing. (**a**) Image grayscale; (**b**) binarization; (**c**) median filtering.

## 4. Parking Space Recognition Algorithm

Parking spaces in parking lots are often divided into two types: parking spaces with a complete rectangular parking space line and an incomplete parking space line. The parking space with a complete rectangular parking space line can be extracted by the Canny edge detection method to extract the parking space line contour, and then the parking space line existing in the contour is detected by the Hough transform. The fake parking space line is eliminated to obtain the parking space information [27]. The parking space with incomplete parking line can be matched with the image to be tested through the pixel gradient template of the parking angle image to recognize the position of the parking angle in the image [28]. In addition, a parking-slot-marking detection approach based on deep learning is proposed. The detection process involves the generation of mask of the marking-points by using the Mask R-CNN algorithm, extracting parking guidelines and parallel lines on the mask using the line segment detection (LSD) to determine the candidate parking slots [29]. However, it is not easy to ensure the recognition accuracy when the parking angle and parking line are worn or blocked. The deep learning recognition algorithm is complex, and a large number of samples need to be collected and data labeled in the experiment, so the processor requirements are high. In this paper, the parking space recognition algorithm is proposed to recognize the angle of the parking space based on locating and identifying the constructed features. To begin with, whether the parking space is partially occupied by the adjacent vehicles is determined. Secondly, the constructed feature that is easy to wear are repaired online through the template matching algorithm. Then, the position of the feature is located and the parking space information carried by the feature is recognized. Finally, the recognition of the parking space and its boundary according to the relative position relationship between the constructed feature and the parking space is realized.

### 4.1. Determination of Parking Space Occupied

After the vehicle automatically drives to the designated parking space according to the recommended route, the vehicles near the parking space may cross the lane and the parking space is partially occupied. A method is designed to identify effective empty parking spaces. On the one hand, the feature contour is extracted after image preprocessing, and the edge of the contour is detected by Canny operator. The purpose of contour extraction is to integrate the pixels of contour edge into a whole [30]. On the other hand, the findContours( ) function in the Opencv library is used to find the contour from the binary image, and the contour through the drawContours( ) function is drawn. As is shown in Figure 4, if the extracted outer contour is not a regular quadrilateral, it means that the parking space is occupied by the adjacent vehicle crossing the line. Later the parking space is redistributed.

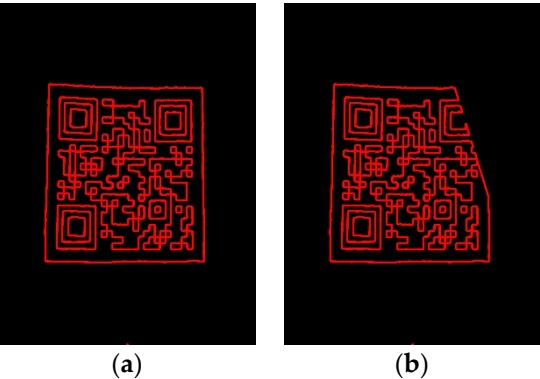

(**a**) (**b**)

**Figure 4.** (**a**) Feature of the free parking space; (**b**) feature of the occupied parking space.

### 4.2. The Template Matching of the Defacement Feature

The QR code itself has the function of error correction. While vehicles enter and exit a parking space frequently, the data information on both sides of the QR code is worn and the vehicle oil dripping causes pollution, resulting the area of contamination beyond

its own error correction range. QR code will not be recognized. Besides, while the key area of the QR code surface is lost or defaced, the defaced area does not exceed the error correction capability of the error correction code, which will lead to the invalid recognition. This paper is based on the template matching method to realize the recognition of the stained QR code, and a new similarity calculation formula is proposed. This strategy is to calculate the similarity between the known image and its template image to find the most similar template feature. Firstly, we extract the pre-processed features that meet the parking conditions, and perform operations such as rotation and shearing, so that the features to be matched are spatially matched with the template. They are matched with all QR code feature templates in the parking lot. Finally, we use the template with the highest similarity as the output. The principle of template matching is to translate the image to be matched in the template. The QR code to be matched is $T$, the template is $M$, their size are all $K \times L$, the standardized binary matrix of the QR code contains only 0 and 1 characters. In order to improve the efficiency and accuracy of matching, a new similarity calculation formula is proposed as follows:

$$S_i = \frac{\sum\limits_{m=1}^{K} \sum\limits_{n=1}^{L} T(m,n) \odot M_i(m,n)}{K \times L} \tag{6}$$

where $T(m,n)$ is the word module corresponding to the $m$ row and $n$ column of the QR code binary matrix to be matched, which represents 0 or 1; $M_i(m,n)$ is the word module corresponding to $m$ rows and $n$ columns of the QR code binary matrix of template $i$, which represents 0 or 1. $\odot$ is the symbol of logic, if the characters are the same, the result is 1; if the characters are different, the result is 0.

The correlation coefficients are obtained by searching the patterns existing in the template library one by one, and the maximum $S$ value was finally selected as the matching target image to realize the online repair of the features of the defaced QR code.

*4.3. The Recognition of Parking Angles*

The location and recognition process of QR code features is shown as follows. Finder patterns are the special position-detection patterns located in three corners (upper left, upper right, and lower left) of each QR code. The detect( ) function is utilized to locate the preprocessed QR code image in the opencv4 library, which can realize the positioning of the QR code at any rotation angle. $A(u_1, v_1)$, $B(u_2, v_2)$, $C(u_3, v_3)$, $D(u_4, v_4)$ are the results of output. They are the pixel coordinates of the four vertices of the quadrangle containing the smallest area of the QR code. After searching for the location of the QR code, the decode( ) function is used to decode the constructed QR code to obtain the parking space features, including the parking space number N. Then the vehicle transmits it to the local server to obtain the parking space occupancy status and the number of remaining parking spaces. The length L and width W of the parking space are transmitted to the ECU for subsequent calculation and processing. In addition, information about whether the parking space is charged or not is acquired. These can realize the information interaction between the parking space and the vehicle.

As shown in Figure 5, $A'(x_1, y_1)$, $B'(x_2, y_2)$, $C'(x_3, y_3)$, $D'(x_4, y_4)$ are the four angles of the parking space. Since the QR code is pasted in the middle of the parking space, the relative position relationship has been determined, and the length and width data of the parking space can be obtained by recognizing the QR code. We perform the similar triangle relationship, the actual distance and pixel distance ratio conversion by using the following formulas:

$$\frac{W-L}{2} \bigg/ \frac{W+L}{2} = \frac{x_1 - u_1}{x_1 - u_4} = \frac{y_1 - v_1}{y_1 - v_4} \tag{7}$$

$$\frac{W-L}{2} \bigg/ \frac{W+L}{2} = \frac{x_1 - u_2}{x_1 - u_3} = \frac{y_1 - v_2}{y_1 - v_3} \tag{8}$$

$$L / \frac{W + L}{2} = \frac{u_2 - u_3}{u_2 - x_4} = \frac{v_2 - v_3}{v_2 - y_4} \tag{9}$$

$$L / \frac{W + L}{2} = \frac{u_1 - u_4}{u_1 - x_4} = \frac{v_1 - v_4}{v_1 - y_4} \tag{10}$$

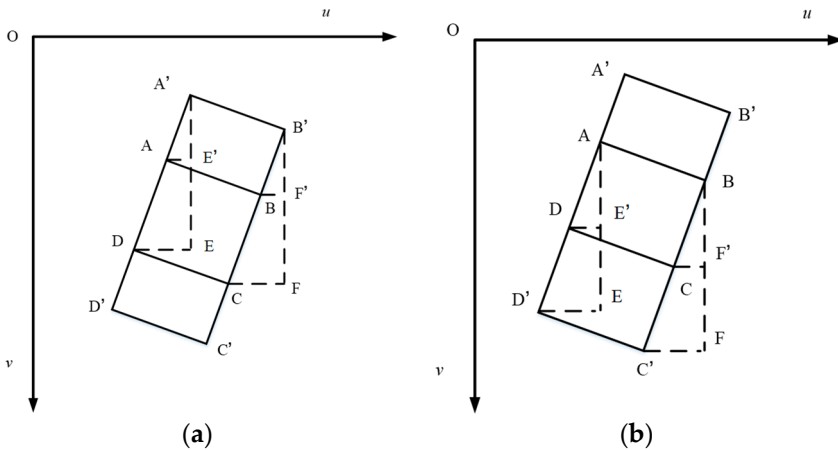

**Figure 5.** (**a**) Similarity transformation of rear angles; (**b**) similarity transformation of front angles.

We can get the pixel coordinates of the four parking angles in the top view:

$$A'(x_1, y_1) = \left( \frac{W \times u_1 + L \times u_1 - W \times u_4 + L \times u_4}{2 \times L}, \frac{W \times v_1 + L \times v_1 - W \times v_4 + L \times v_4}{2 \times L} \right) \tag{11}$$

$$B'(x_2, y_2) = \left( \frac{W \times u_2 + L \times u_2 - W \times u_3 + L \times u_3}{2 \times L}, \frac{W \times v_2 + L \times v_2 - W \times v_3 + L \times v_3}{2 \times L} \right) \tag{12}$$

$$C'(x_3, y_3) = \left( \frac{L \times u_2 - W \times u_2 + W \times u_3 + L \times u_3}{2 \times L}, \frac{L \times v_2 - W \times v_2 + W \times v_3 + L \times v_3}{2 \times L} \right) \tag{13}$$

$$D'(x_4, y_4) = \left( \frac{L \times u_1 - W \times u_1 + W \times u_4 + L \times u_4}{2 \times L}, \frac{L \times v_1 - W \times v_1 + W \times v_4 + L \times v_4}{2 \times L} \right) \tag{14}$$

Four parking angles are successively connected by straight lines, and the obtained outer rectangle is displayed in the original parking image to realize the identification of free parking spaces.

## 5. Experiments

The experiment was made up of the hardware parts, as shown in Figure 6, and the software parts. The hardware consisted of fisheye cameras (FE185C057HA-1, Fujifilm, Tokyo, Japan, 2020) that are installed on the center of the front and rear bumpers of the vehicle, also on the left and right side of the mirrors. The fisheye cameras on the left and right were used for detection and recognition of the parking space, and the fisheye cameras on the front and rear were used for the vehicle to identify potential front and rear obstacles. While the software part consisted of image processing by the on-board computer. The implementation of the software part was done using C++ with Visual Studio (VS2019, Microsoft, Redmond, WA, USA, 2019) as the development environment using OpenCV library (Version 4.1, Intel, Santa Clara, CA, USA, 2019) for image processing.

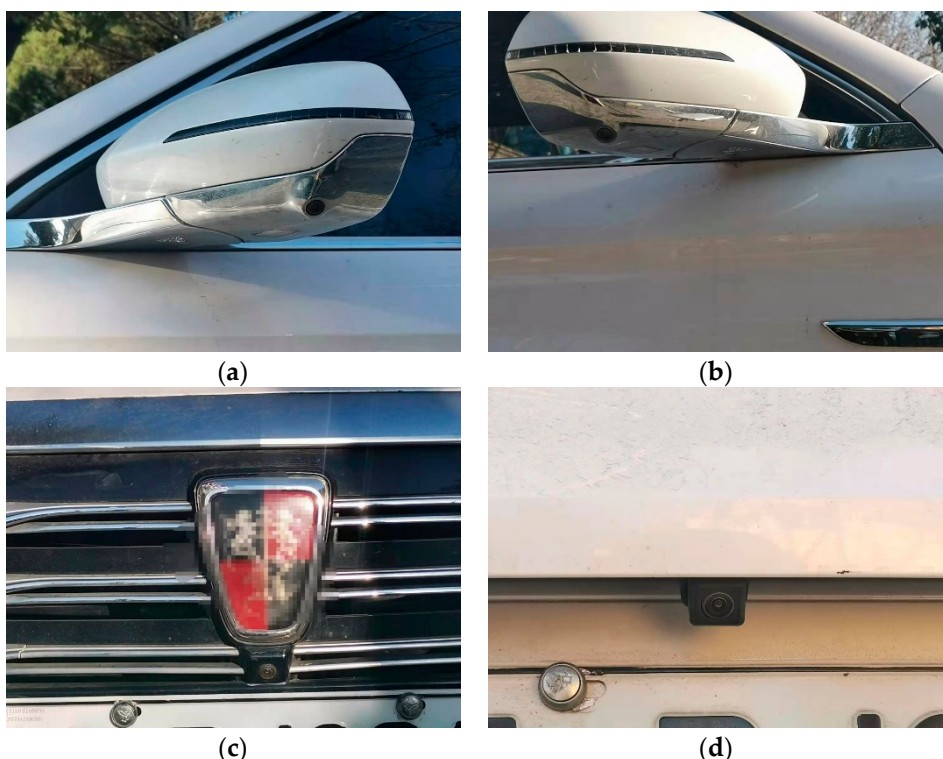

**Figure 6.** (**a**) Fisheye cameras on the left; (**b**) fisheye cameras on the right; (**c**) fisheye cameras on the front; (**d**) fisheye cameras on the rear.

In order to verify the effectiveness of the parking space recognition method based on parking space features, 120 QR codes with version number 2 and error correction level H are selected as the template library. Each of them represents the feature of each parking space in the parking lot. After we performing distortion correction and top view transformation on the worn QR code features captured by the fisheye camera, and we perform operations such as rotation and shearing so that the features to be matched are spatially matched with the template. Then we write C++ code to calculate the similarity between the stained QR code and the template in the template library, and select the one with the highest similarity as the output result. The maximum similarity value in this experiment is 89.66%, as shown in Figure 7.

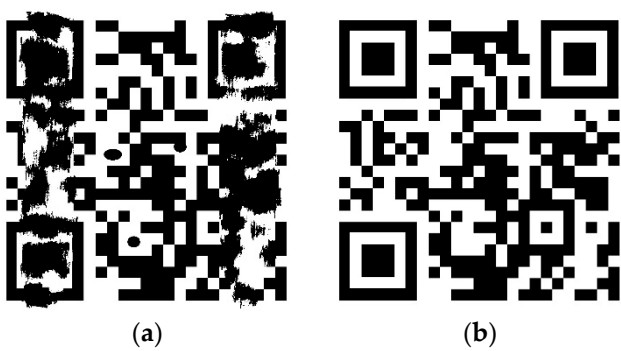

**Figure 7.** (**a**) Defaced QR code; (**b**) matched QR code.

The results of traditional Hough transform and parking angle recognition based on template matching are shown in Figure 8. As shown in Figure 8a, the traditional Hough transform is susceptible to the interference of a straight line in the background environment, and the left parking space line is not detected due to partial wear. As shown in Figure 8b,

the parking angle recognition method based on template matching requires the parking space angle to be complete. Once the parking space angle is worn, it cannot be recognized.

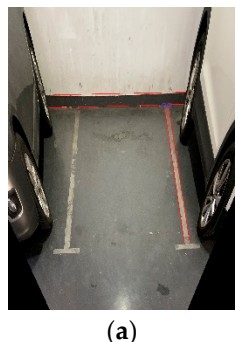
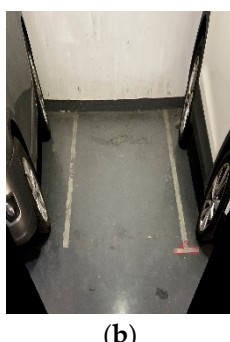

(**a**)            (**b**)

**Figure 8.** Recognition results of parking spaces. (**a**) Parking line recognition method based on Hough transform; (**b**) parking angle recognition method based on template matching.

In order to verify whether the intensity of light will affect the recognition of different types of parking spaces in the actual parking environment, an experiment was carried out. The results are shown in Figure 9. Nevertheless, when the vehicles near the parking space cross the parking lines and the parking space is partially occupied, it was a challenge to identify invalid occupied parking spaces. However, when we experimented with the proposed method of extracting outer contour, the invalid occupied parking spaces could easily be identified, as indicated in Figure 10.

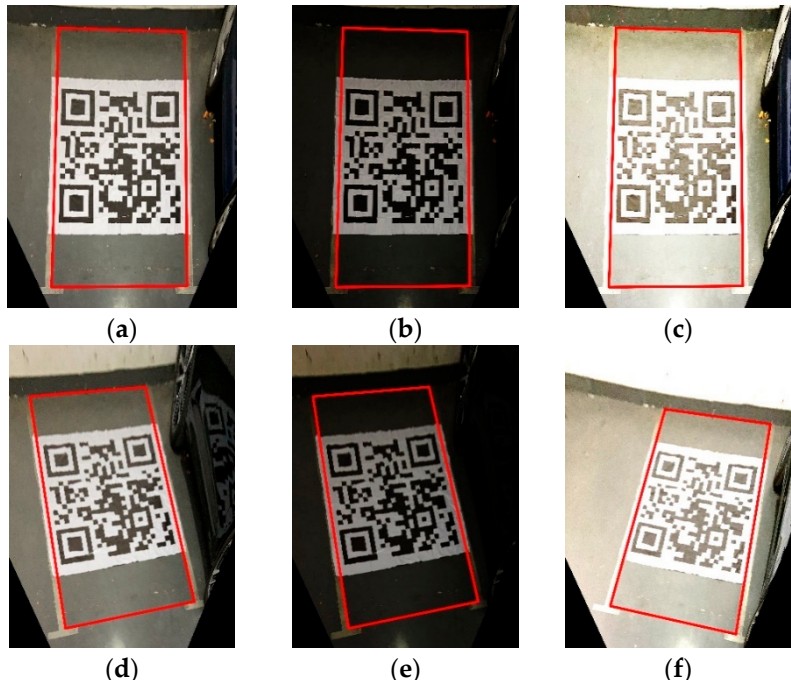

(**a**)            (**b**)            (**c**)

(**d**)            (**e**)            (**f**)

**Figure 9.** Recognition results of effective parking spaces. (**a**) Vertical parking space with normal light; (**b**) vertical parking sapce with low light; (**c**) vertical parking space with strong light; (**d**) tilt parking space with normal light; (**e**) tilt parking space with low light; (**f**) tilt parking space with strong light.

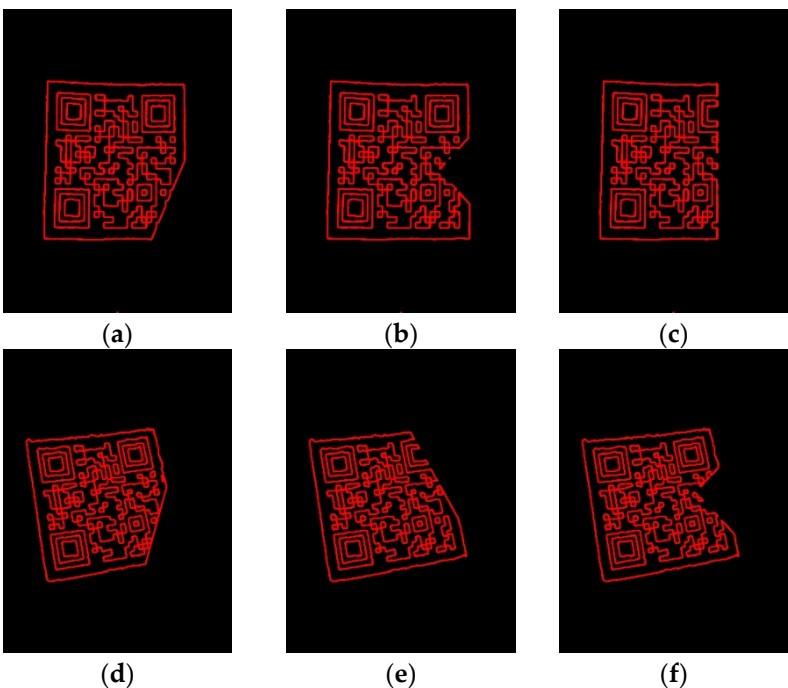

**Figure 10.** Recognition results of occupied parking spaces. (**a**) Occupied vertical parking space 1; (**b**) occupied vertical parking space 2; (**c**) occupied vertical parking space 3; (**d**) occupied tilt parking space 1; (**e**) occupied tilt parking space 2; (**f**) occupied tilt parking space 3.

Table 1 shows the comparison of parking space recognition results.

**Table 1.** The result of comparison of parking space recognition results.

| Method | Number of Samples | Number of Positive Samples | Precision | Accuracy | Recall | F1 |
|---|---|---|---|---|---|---|
| New algorithm proposed | 120 | 90 | 95.45% | 70% | 93.33% | 0.94 |
| Parking line recognition method based on Hough transform | 120 | 90 | 86.67% | 54.17% | 72.22% | 0.79 |
| Parking angle recognition method based on template matching | 120 | 90 | 85% | 56.67% | 75.56% | 0.8 |

## 6. Conclusions and Future Research

In order to the deal with the challenge existing in current autonomous valet parking, this paper presents a method to recognize parking spaces based on parking space feature construction. The cameras are used to extract features' contour, locate features' position, and recognize features. The parking spaces and their boundary are identified through the relative position relationship between the constructed feature and the parking space. In addition, this method not only judges the occupancy of parking spaces, but also forms closed-loop communication among the parking lot and parking space and vehicle. The experimental results showed that this method could effectively identify different types of parking spaces in various conditions. For future research, more information communication will be taken into consideration between the constructed feature and the vehicle.

**Author Contributions:** S.M. and W.F. designed the scheme. H.J., M.H. and C.L. checked the feasibility of the scheme. S.M. and H.J. provided the resources. W.F. performed the code and experiment. W.F. wrote the paper with the help of S.M. All authors have read and agreed to the published version of the manuscript.

**Funding:** This research was supported by the Natural Science Fund for Colleges and Universities in Jiangsu Province under Grant 12KJD580002 and Grant 16KJA580001, in part by the Innovation Plan for Postgraduate Research of Jiangsu Province in 2004 under Grant KYLX1057, and in part by the National Natural Science Foundation of China under Grant 51675235.

**Institutional Review Board Statement:** Not applicable.

**Informed Consent Statement:** Not applicable.

**Data Availability Statement:** The data presented in this study are available on request from the corresponding author.

**Conflicts of Interest:** The authors declare no conflict of interest.

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
