# Peer review of "Parking Space Recognition Method Based on Parking Space Feature Construction in the Scene of Autonomous Valet Parking"

_applsci, doi:10.3390/app11062759_

Round 1
Reviewer 1 Report
I. General remarks. 1. An article: Parking space recognition method based on parking space feature construction in the scene of autonomous valet parking is an interesting, strictly practical paper and it is suitable for publication in this ( applied science) journal. In this work, I did not find much from the field of science, but applied is sufficiently represented. The work is devoted to a combination of pattern recognition (QR codes) with image processing (distortion correction and invers perspective transformation) for parking space recognition. 1. It is also not clear why the car has been equipped with two cameras located on the side mirrors? If they are really useful, please specify when and how the information from the three cameras is aggregated. For front parking - the front and two sides cameras, when parking backwards - the rear and two sides cameras. 2. In this work ( rightly referred as applied), there is a lack of tests of the proposed method, on real life data sets, and this is the basic shortcoming of this work. That, what is included in Chapter 5 Experiments, is far from the standards used for testing, in applied works. The standard is to relate the obtained results to the achievements of other methods already used. Then precision, accuracy, recall or F1 metrics can be compared. 3. The proposed methodology for assessing the proposed solution is a little “exotic”. Based on test data sets of 500 QR codes (see page 9), Authors show the results for 220 QR codes (see Table 1, page 11). Please, explain for the readers: What was rationale, that only 220, from 500, have been chosen? What was the selection criterion? From what real life conditions of the experiment this criterion results and how it has been influence the obtained results? 4. Proposed approach has further implications; proposed in the paper, recognition rate metrics (Table 1, page 11) limits accuracy only to percent of the total number of samples that are correctly classified. Considering the standard confusion matrix, we have Precision taking into account not only (like Purity) correctly classified (true positive) but also false positive. Next, we have Recall and F1 metrics, so such broad perspective allows to evaluate the relevance of the results. 5. Please refer to other solutions used (e.g. Bosh, Mercedes, Continental, Daimler) and show why it isn’t good enough? and why new, proposed in this paper, solution is better than other approaches known in automotive technology II. To ensure a high level presentation of obtained results I have the following detailed comments. 1. Page 3, lines 111-112, “…multi-dimensional information,…” Please specify, what kind of multidimensional information is stored in QR code? The parking spot, that the car will aim for, is selected by AVP system to suit the vehicle's size. Also the route to a free parking spot is computed by parking garage infrastructure and by digital transmission passed to the car. Provide the rationale of proposed approach, please. 2. Page 3, line 150, “… 3.1. Image acquisition.” This chapter describes two image pre-processing methods such as distortion correction (see lines 155, 161, 163, and Fig.1 ) and inverse perspective transformation (see lines 169, 172, formula(1) and Fig.2). Therefore these descriptions should be transferred to the next chapter, 3.2. Image preprocessing. Amend this, please! In this (3.1) chapter, please describe how images from three cameras are acquired and how these images are aggregated into one pre-processed image. 3. All formulas in the paper; (typographic alignment) Please centre formulas and justify the numbers in brackets to the right side ! 4. Page 5, formula(1); Why such factors have been chosen for RGB components? The proposed values correspond to the perception of the human eye. CIE perceptual models of the colour space; xyz and lab, created for the European population, have different coefficients, as those created for the American population. Can they be applied to the camera? What will be the consequences of doing so? 5. Page 5, Figure.3, Fig. 3b; to what value and how the binarization threshold was determined? Is it a constant or is it a variable being a function of lighting conditions? If the latter, then provide mathematical formula of this function, please! Provide the rationale of such choice, please! If we need black and white images, maybe we should not use colour cameras and just install black and white cameras? The proposed solution is like buying a Lamborghini Huracán EVO (640 hp) and then updating the LDS software to reduce traction power to 100 hp! Does it make sense? Fig. 3c; Comparing figures. 3b with 3c, it is difficult to see significant differences. So, could we resign from median filtering that does not improve the image but introduces noise, because smoothing and blurring are such. How were the parameters of these two processes (smoothing, blurring) determined? Provide the rationale of such choice, please! 6. Page 5, lines 206-209; “…In addition, a parking-slot-marking detection approach based on deep learning is proposed. The detection process involves the generation of mask of the marking-points by using the Mask R-CNN algorithm, extracting parking guidelines and parallel lines on the mask using the line segment detection (LSD) to determine …” Is this not art for art's sake? In existing AVP systems (e.g. Bosh, Daimler), such software is on the garage side (ecomics and logic!), Because there are fewer of them than cars). Moreover, car cameras only see their neighborhood. How long will the car be waltzing to get close enough to the parking spot and decide if it can fit to park here? Existing AVP systems provide such information within the drop-off area of the parking garage and the car "knows" where to go. Also, if this software will run on a car, what is the point of using deep learning? The R-CNN algorithm will average situations from different garages, different light conditions, etc. How can such an averaged value be useful in a specific garage and specific situation? One final note; why the QR codes were introduced if we now return to the known (and successfully used line segment detection solution (see line 209))? 7. Page 6, Figure 4b; Very sophisticated method to realize very simple tasks! If the QR code shape is not a square, then we don't park in this spot. Consequently, we don't read it, because it doesn't matter! 8. Page 6, line 241; “… invalid recognization.” Please, amend to: invalid recognition 9. Page 6, line 247; ; “…with the templat.” Please, amend to: with the template 10. Page 7, formula (3); Please define the operation in the counter (a circle with a point in the center). Some elements of T (m, n) are invisible. What value do we then substitute into formula (3)? 11. Page 7, line 255; “…T(m,n) is the character.” Please, consider to amend to the word module. 12. Page 9, line 308; “…different parking space information …” Please specify exactly elements creating this information set. 13. Page 9, line 311; ; “…with the templat.” Please, amend to: with the template 14. Page 11, lines 334-335; “…but also forms closed-loop communication among the parking lot and parking space and vehicle …” Nowhere at work, have I found anything, that says about such communication, provided by proposed method. 15. Page 12, line 340; “ … performed the software simulation and experiment. …” I have read a lot about the experiment at work (chapter 5), thinking that it was carried out in real life. So, please underline clearly, where and what was simulated and what was a real life experiment. Hopefully, that these remarks allow to improve this work,

Reviewer 2 Report
Interesting paper on automatic valet parking using a relatively simple method based on a QR code incorporated at each parking spot. The method is clearly presented. Preliminary results are presented. The results of a more thorough experimental testing of the method will be interesting - hopefully to appear at a later publication.
Proposed small revisions:
The paper would benefit from a separate Discussion session discussing the advantages and limitations of the method, and comparing it to other methods. Some questions this reviewer has, and I think readers may also have, are: a) how does this method scale, b) discuss QR installation and maintenance requirements.
Regarding presentation, the article will benefit from some proof-reading by a native English speaker. Especially the Abstract appears written in a hurry and requires spell-checking and some editing.
Round 2
Reviewer 1 Report
I have no comments